# Divergent Effects of Resveratrol on Rat Cardiac Fibroblasts and Cardiomyocytes

**DOI:** 10.3390/molecules24142604

**Published:** 2019-07-17

**Authors:** Xavier Lieben Louis, Zach Meikle, Laura Chan, Garret DeGagne, Rebecca Cummer, Shannon Meikle, Sampath Krishnan, Liping Yu, Thomas Netticadan, Jeffrey T. Wigle

**Affiliations:** 1Institute of Cardiovascular Sciences, St. Boniface Hospital Albrechtsen Research Centre, Winnipeg, MB R2H 2A6, Canada; 2Department of Biochemistry and Medical Genetics, University of Manitoba, Winnipeg, MB R3E 0J9, Canada; 3Canadian Centre for Agri-Food Research in Health and Medicine, St. Boniface Hospital Albrechtsen Research Centre, Winnipeg, MB R2H 2A6, Canada; 4Agriculture and Agri-Food Canada, Winnipeg Winnipeg, MB R2H 2A6, Canada; 5Department of Physiology and Pathophysiology, University of Manitoba, Winnipeg, MB R3E 0J9, Canada

**Keywords:** cardiac fibrosis, resveratrol, cardiac fibroblasts, cardiomyocytes

## Abstract

In this study, we tested the potential cardioprotective effects of the phytoalexin resveratrol (Rsv) on primary adult rat cardiac fibroblasts (CF), myofibroblasts (MF) and cardiomyocytes. Adult rat CF and cardiomyocytes were isolated from male 10-week old Sprague–Dawley rats, cultured for either 24 h (cardiomyocytes) or 48 h (CF) before treatments. To isolate MF, CF were trypsinized after 48 h in culture, seeded in fresh plates and cultured for 24 h prior to treatment. All three cells were then treated for a further 24 h with a range of Rsv doses. In CF and MF, cell proliferation, viability, apoptosis assays were performed with or without Rsv treatment for 24 h. In cardiomyocytes, cell viability and apoptosis assay were performed 24 h after treatment. In separate experiments, CF was pre-incubated with estrogen, tamoxifen and fulvestrant for 30 min prior to Rsv treatment. Rsv treatment decreased proliferation of both fibroblasts and myofibroblasts. Rsv treatment also increased the proportion of dead CF and MF in a dose dependent manner. However, treatment with Rsv did not induce cell death in adult cardiomyocytes. There was an increase in the percentage of cells with condensed nuclei with Rsv treatment in both CF and MF, but not in cardiomyocytes. Treatment with estrogen, tamoxifen and fulvestrant alone or in combination with Rsv did not have any additional effects on CF survival. Our results demonstrate that treatment with Rsv can inhibit cell proliferation and induce cell death in rat CF and MF, while not affecting cardiomyocyte survival. We also demonstrated that the induction of cell death in CF with Rsv treatment was independent of estrogen receptor alpha (ERα) signaling.

## 1. Introduction

Heart failure (HF) is the end stage clinical manifestation of heart disease [1]. HF is a growing epidemic in Canada, with 50,000 new cases diagnosed each year [2]. Cardiac fibroblasts and cardiomyocytes are two cell types that play major roles in normal cardiac function, as well as in HF [3,4,5]. During a myocardial infarction (MI), cardiomyocytes die due to a lack of blood supply [6]. After cardiomyocyte death and the subsequent inflammatory response to clear damaged cells and debris, resident cardiac fibroblasts (CF) are activated and initiate the wound healing process [7]. Activated resident CF convert into myofibroblasts (MF) that secrete elevated levels of extracellular matrix (ECM) proteins to maintain the structural integrity of the heart, which was compromised due to loss of cardiomyocytes from MI [8]. Prolonged activation of MF leads to excessive ECM secretion in the remaining healthy regions of the heart, causing stiffening of the myocardium and reduced heart function, which further progresses into heart failure [3].

Nutraceuticals, the component of food that provides health benefits, have been increasingly explored as supplementary therapies for heart disease [9,10]. Recently, the polyphenol, resveratrol (Rsv), has been investigated as a possible cardioprotective nutraceutical [11]. Rsv is found in the skin of grapes, blueberries, raspberries and mulberries [10]. We previously published a review that discussed studies reporting beneficial effects of Rsv in preventing and reversing abnormalities in cardiac function and structure [12]. Rsv is also a phytoestrogen, a class of compounds with a structure similar to that of estrogen (E2). It has been suggested that the cardioprotective properties of Rsv may be mediated through E2 signaling [13]. Moreover, it is intriguing that Rsv can induce cell death in some cell populations while preventing cell death in other cell types [14]. This cell type specific mechanism of action is potentially a beneficial property for a cardiovascular drug as it may prevent cardiomyocyte death while inducing cell death in proliferating CFs and MFs after cardiac injury. In this study, we compared the effects of Rsv in vitro on the survival of primary rat CFs, MFs, and cardiomyocytes. We also examined the effect of Rsv on MF and CF proliferation.

## 2. Results

### 2.1. Comparing the Dose Dependent Effect of Resveratrol on Fibroblast, Myofibroblast and Cardiomyocyte Cell Death

Cell count analyses of red and green stained cells representing live and dead cells, respectively, showed that Rsv treatment increased the percentage of dead cells in both CF and MF in a dose dependent manner (Figure 1A–C). The 30 and 60 µM doses of Rsv significantly increased cell death in CF and MF. There was no difference in the percentage of dead cells between Rsv 30 and 60 µM doses. The percentage of dead cells increased by 50% with Rsv treatment in both CF and MF. There was no difference in the percentage of dead cells between untreated MF and CF (Figure 1C). In cardiomyocytes, dead cardiomyocytes (red) were counted by considering cell morphology. Rsv treatment did not alter the rate of cardiomyocyte cell death (Figure 1C). Treated or untreated cardiomyocytes had a significantly lower percentage of dead cells as compared to CF and MF (Figure 1C). Among the cardiomyocytes, the small red dots (significantly smaller and morphologically different from dead cardiomyocytes) observed are likely cellular debris or fragments, which were counted separately; no differences were observed between control and Rsv treated groups (data not presented).

### 2.2. Quantification of CF and MF Proliferation

Analyses of cell proliferation using the Cytoquant assay showed that treatment with Rsv decreased total nucleic acid content in both CF and MF, as demonstrated by decreased fluorescent intensity measurements with Rsv treatment in both cell types (Figure 2A). The decrease in intensity was 46% and 42% for CFs and MFs, respectively. Basal fluorescent intensity in untreated MF was lower than the untreated CF, suggesting that the rate of proliferation may slow down as the fibroblasts phenoconvert into MF (Figure 2A).

To further study the effect of Rsv on cellular proliferation, a BrdU incorporation assay was performed. This experiment showed that incubation with Rsv significantly decreased MF proliferation, when compared to vehicle treated control cells (Figure 2B). There was a 58% reduction in MF proliferation with Rsv treatment.

### 2.3. Effect of Rsv on Fibroblasts and Cardiomyocyte Apoptosis

Hoechst staining showed that Rsv treatment significantly increased the proportion of CFs and MFs with condensed, apoptotic nuclei (Figure 3A,B). Rsv treatment increased the percentage of apoptotic cells with condensed nuclei by 72% and 62% in CF and MF, respectively (Figure 3D). In contrast, Rsv treatment did not increase the number of cardiomyocytes with condensed, apoptotic nuclei (Figure 3C,D).

### 2.4. Effect of Estradiol, Tamoxifen and Fulvestrant on Cardiac Fibroblasts

Treatment with E2 did not affect CF cell survival when compared to untreated controls (Figure 4A). Co-incubating CF with E2 and Rsv also did not have any added effect when compared to cells treated with Rsv alone (Figure 4B). Incubation with the estrogen receptor modulator Tmx or the estrogen receptor alpha (ERα) specific antagonist fulvestrant alone did not induce any changes in CF survival, and the addition of Tmx or fulvestrant prior to Rsv treatment did not alter the rate of cell death as compared to that induced by Rsv alone (Figure 4B,C).

## 3. Discussion

The proliferation of CF and their conversion into MF in response to pressure overload or ischemia reperfusion injury is a key pathological step in cardiac fibrosis and a contributor to development of HF [3,6]. There is no effective therapy against cardiac fibrosis, and this has been a hurdle in developing cardiovascular drugs that inhibit the progression of heart diseases into HF [15]. Blocking the angiotensin II signaling that stimulates MF activation was considered a potential therapeutic target [16]. However, anti-fibrotic drugs belonging to the class of angiotensin receptor blockers were hindered by the development of adverse side effects [17]. These challenges highlight the importance of natural polyphenolic molecules like Rsv with potential cardioprotective properties. Rsv has been demonstrated to be well tolerated by older and sicker populations, who are more at risk to any possible side effects of drugs and also more at risk of developing HF [18,19,20]. To date, Rsv treatment has been shown to have minimal side effects often only observed at very high concentrations, which makes it an ideal candidate to test for anti-fibrotic properties [21]. In this study, we found that Rsv induced cell death in both CF and MF in a dose dependent manner. Rsv also increased the rate of apoptotic cell death in both CF and MF. However, Rsv did not induce adult cardiomyocyte cell death or apoptosis at any doses tested in this study. We also demonstrated that Rsv reduced CF and MF cellular proliferation. In this study, addition of estrogen or inhibition of estrogen signaling by Tmx (Selective ER modulator) or Fulvestrant (ERα specific antagonist) receptor had no effect on the action of Rsv on fibroblasts.

We have previously reported that Rsv protects adult rat cardiomyocytes from oxidative stress and norepinephrine induced damages in vitro [22,23] and protects the heart in vivo [13,24,25,26,27]. In the in vivo studies, we showed an improvement in systolic and diastolic function of the heart, which suggests that Rsv treatment may have helped maintain cardiac tissue elasticity, possibly through restricting cardiac fibrosis [8]. Ameliorating the pathophysiological process of fibrosis and subsequent decrease in collagen deposition was reported to be associated with Rsv mediated cardioprotection [28]. The data from this study using adult rat cardiac fibroblasts confirms these in vivo findings by showing that Rsv inhibits the proliferation of both CF and MF. In this study we showed that Rsv treatment increased the rate of cell death in both CF and MF. The resident cardiac fibroblast population has a role in the normal physiological function and removing those cells could affect normal function of the heart. In this study, we observed that Rsv induce cell death in CF. However, these cells are actively proliferating and becoming activated in culture. Moreover, it has been shown that Rsv reduces fibrosis in the pathological heart, but normal rat heart functions were unaffected in control animals [24]. This suggests that Rsv may preferentially target the MF population in vivo. Another concern would be if Rsv induces cell death in other cardiac cell types. In this study, we observed that when treated with the same doses of Rsv, cardiomyocyte cell survival was unaffected. This observation is consistent with our previous report that Rsv protected neonatal cardiomyocytes against norepinephrine induced apoptosis, while inducing apoptosis in proliferating cardiac tumor cells [14]. Rsv induces cell cycle arrest, down regulates cellular survival mechanisms, disrupts mitochondrial function and induces apoptotic signaling to trigger cell death in highly proliferative, transformed cancer cells [29]. In contrast, Rsv promotes cell survival through a different mechanism involving decreasing oxidative stress, inflammation, improving mitochondrial function and activating anti-apoptotic mechanisms [30,31]. This is a therapeutically relevant property of Rsv that, at the same concentration, Rsv can alter cell survival differently according to specific cell type being studied. We observed an increase in the percentage of cells with condensed nuclei (a hallmark of apoptosis) in CF and MF but not in cardiomyocytes. This suggests that Rsv induces cell death in CF and MF through apoptosis. Rsv treatment reduced cellular proliferation of both CF and MF. Therefore, our data suggest that Rsv may limit the pathological effects of cardiac fibrosis by limiting CF/MF expansion through a combination of decreasing proliferation and inducing cell death.

Rsv has been shown to function via ERα in neural and vascular smooth muscle cells [32,33] and Rsv attenuates cardiac dysfunction due to E2 deficiency [34]. Similar to E2, Rsv activates downstream targets like 5’ AMP-activated protein kinase (AMPK), endothelial nitric oxide synthase (eNOS), NAD-dependent deacetylase sirtuin-1 (SIRT1) and antioxidant enzymes, suggesting that they share a common signaling pathway [35]. There is strong evidence suggesting that the phytoestrogenic property of Rsv is a major contributor to its cardioprotective effects [13]. Several studies have shown that Rsv mediated cardiovascular protection is mediated through upregulating ERα or by activating downstream proteins in ERα signaling pathway [36,37,38]. Comparatively, in this study, activation of ERα using E2 alone did not alter CF death. There was also no synergistic effect when E2 was co-incubated with Rsv, suggesting a role for a non-ER mediated signaling in CF death induced by Rsv. This was further supported by using Tmx that blocks both ERα and ERβ prior to Rsv addition. We also used Fulvestrant, which is a selective estrogen receptor degrader that has higher affinity to ERα receptor. Pre-incubation with Tmx or fulvestrant had no effect on Rsv mediated CF death suggesting an ER independent signaling with Rsv. Our finding is consistent with an earlier study using CFs and showed that Rsv treatment did not affect protein kinase B and ribosomal protein S6 kinase beta-1 (Akt/p70S6K) activity, which are downstream targets of ER signaling [39]. The same study showed that Rsv inhibited CF proliferation induced by angiotensin II through inhibition of mitogen-activated protein kinase (MAPK) signaling. Accordingly, we suggest that Rsv may be acting on CF by inhibiting growth signaling, rather than inducing cell death through the classical ER pathway.

In conclusion, this study shows that in adult rat CF and MF, Rsv reduces proliferation and induces cell death, while adult rat cardiomyocyte viability was unaffected by Rsv treatment. Rsv mediated CF death was independent of ER-alpha signaling. Further studies are required to elucidate the exact mechanism of action of Rsv on CF. This study reinforces the potential of Rsv as a cardioprotective agent that could possibly be used in the future as a supplementary therapy against cardiac diseases involving cardiac fibrosis.

## 4. Materials and Methods

The experimental protocol used in this project were approved (protocol number 17-003) by the University of Manitoba Office of Research Ethics and Compliance and Animal Care Committee and were conducted in accordance with guidelines by the Canadian Council for Animal Care.

### 4.1. Adult Cardiomyocyte and Fibroblast Isolation and Culture

All chemicals used for primary cell isolation were purchased from Sigma-Aldrich (Oakville, ON, Canada). Adult cardiomyocytes and fibroblasts were isolated from 10-week old male Sprague–Dawley rats weighing around 250–300 g from the University of Manitoba animal breeding facility. Briefly, rats were anesthetized using ketamine/xylazine (90/10 mg/kg body weight) administration. Heparin (1000 USP units/kg body weight) was administered intravenously through the left femoral artery. The heart was then excised and transferred to ice cold saline solution. The heart was rinsed with saline solution and mounted on the Langendorff perfusion apparatus. The heart was perfused with calcium-free buffer containing (in mmol/L) NaCl 90, KCl 10, KH_2_PO_4_ 1.2, MgSO_4_ · 7H_2_O 5.0, NaHCO_3_ 15, taurine 30, glucose 20, pH 7.4 and collagenase (0.5 mg/mL) for 40 min. The ventricles were dissected into 8–10 pieces and incubated in 10 mL Ca^2+^ free buffer collected from the Langendorff system, in 37 °C water bath for 10 min. Cells were separated from the tissue by repeated pipetting using a 5 mL pipette. The supernatant was transferred into a fresh 50 mL tube. The cells collected in the 50 mL tube were left undisturbed for 5 min and then the supernatant containing fibroblasts were transferred into a separate 50 mL tube. This step was repeated 4–5 times. The sedimented cells were used for cardiomyocyte experiments as described earlier [22]. A penicillin/streptomycin (Gibco, Life Technologies, ON, Canada) antibiotic mixture was added to tube containing fibroblasts and was centrifuged at 2000 rpm for 5 min in room temperature. The pellet was re-suspended in sterile phosphate buffer saline (PBS) with penicillin/streptomycin antibiotic mixture and centrifuged again at 2000 rpm for 5 min in room temperature. The pellet was re-suspended in DMEM-F12 medium (ThermoFisher Scientific, Ottawa, ON, Canada) with 10% fetal bovine serum (FBS; SigmaAldrich, Oakville, ON, Canada) and penicillin/streptomycin antibiotic mixture, plated 2 mL/well into 6 well plates and incubated at 37 °C in a CO_2_ incubator. The cardiomyocytes were plated onto 6 well plates after transferring through increasing concentrations of Ca^2+^ and finally suspending in M199 medium (ThermoFisher Scientific, Ottawa, ON, Canada) with 10% FBS. After 2–3 h, CF were washed two times with sterile PBS and incubated in fresh DMEM-F12 medium supplemented with 10% FBS until the day of experimentation. For cardiomyocytes, the medium was replaced with fresh serum free M199 medium supplemented with 5 mM taurine, 2 mM carnitine, 1 mM creatine and 1 μmol insulin, and incubated until experimentation. After 48 h, CF were either treated as outlined below or treated with trypsin and seeded onto new plates at 1:2 ratio and maintained in DMEM-F12 medium with 10% FBS overnight. These cells were considered as myofibroblasts (MF) after 72 h of culture post-isolation (or 24 h after re-seeding) [40,41].

### 4.2. Dose Dependent Effect of Resveratrol on Cardiac Fibroblasts, Myofibroblasts and Cardiomyocytes

Rsv was dissolved in 100% dimethyl sulfoxide (DMSO) and then diluted with sterile water to prepare a 50% DMSO solution at 60 mM stock concentration. CF or MF medium with 10% serum was replaced with serum free media 3 h before the Rsv treatment. Cells were maintained in serum free medium until the end of the experiment. CF, MF and cardiomyocytes were treated for 24 h with a range of Rsv doses (5, 10, 30 and 60 µM) or vehicle control. Live dead cell assay was used to determine the effect of resveratrol on CF, MF and cardiomyocytes. Briefly, 24 h after treatment with Rsv, CM, MF and cardiomyocytes were incubated with calcein AM (Invitrogen, ON, Canada) and ethidium homodimer 1 (Biotium, CA, USA) for 20 min. Ethidium homodimer 1 is highly positively charged and hence non-permeable in live cells. It enters cells with damaged membrane and emits red fluorescence when bound to DNA. Calcein AM is a cell permeable dye that gets converted to its fluorescent form after hydrolysis by intracellular esterases and is used to detect viable cells. Red and green cells were imaged using EVOSfl fluorescent microscope (AMG, ThermoFisher Scientific, MA, USA) at a 10× magnification. The images were analyzed using ImageJ software. All analyses were done in a blinded manner.

### 4.3. Quantification of CF and MF Proliferation

To examine effect of Rsv on CF and MF cell growth, we used a CyQUANT Direct Cell Proliferation Assay kit (Invitrogen, Ottawa, ON, Canada) according to manufacturer’s protocol. CF and MF were incubated with CyQUANT nucleic acid stain and background suppressor solution for 30 min at the end of the 24 h treatment. For CF, the cell count was not determined before seeding because the initial population of cells are heterogeneous. Comparable numbers of cells were plated in each well by thoroughly mixing the cell solution before seeding. Seeding density was visually confirmed before treating the cells with Rsv. In contrast to other cell types, fibroblasts are not washed off during medium change (24 h post isolation) and proliferate rapidly resulting in a homogenous population at 48 h. Fluorescence intensity of nucleic acid staining was read at 485/530 nm (excitation/emission) using a Cytation 5 imaging reader (BioTek, VT, USA). An increase or decrease in fluorescence intensity is directly proportional to the DNA content which in turn is directly proportional to the total number of live cells.

To measure the effect of Rsv on MF proliferation, fibroblasts were trypsinized 48 h after isolation and reseeded at a density of 1 × 10^5^ cells per well in a 96 well plate in DMEM-F12 medium with 10% serum; after 24 h bromodeoxyuridine 5-bromo-2′-deoxyuridine (BrdU) cell proliferation assay kit (Millipore (Canada) Ltd., Etobicoke, ON, Canada) was used to measure MF proliferation. BrdU label was added with Rsv (30 µM) in DMEM-F12 medium without serum and incubated with the cells for further 24 h. At 96 h after isolation, cells (MF) were fixed and processed according to the manufacturer’s protocol (Millipore (Canada) Ltd., Etobicoke, ON, Canada). Two negative controls were used for this assay, (1) without cells but with BrdU label; and (2) with cells, but no BrdU label.

### 4.4. Measurement of Apoptotic Cells

Hoechst assay was used to detect apoptotic cells in CF, MF and cardiomyocytes. Twenty-four hours after Rsv treatment, CF, MF and cardiomyocytes were incubated with 1.5 µL/well Hoechst 3342 dye (10 mg/mL; ThermoFisher, Ottawa, ON, Canada) for 20 min in CO_2_ incubator. Cells were then imaged using EVOSfl fluorescent microscope at 10X magnification. The images were analyzed using ImageJ software.

### 4.5. Fibroblast Treatment with Estrogen, Tamoxifen and Fulvestrant

Estrogen, tamoxifen (Tmx) and fulvestrant were dissolved in 100% DMSO at a concentration of 1 mM. Forty-eight hours after isolation, in separate experiments, CF were either treated with 1 µM E2, (Cayman Chemical, MI, USA) or 250 µM Tmx (Cayman Chemical, MI, USA) or 100 nM fulvestrant (Sigma Aldrich, ON, Canada) for 30 min before incubating with Rsv for 24 h. The live and dead assay was performed after the 24 h of treatment as described earlier.

### 4.6. Statistics

Prism Graphpad software (version 5.0) was used for all statistical analyses used in this study. Two-Way-ANOVA was used to analyze dose dependent cell death data. Two tailed student t-test was used to analyze BrdU cell proliferation assay data. All other data were analyzed by one-way ANOVA with Tukey post-hoc test. P value less than 0.05 was considered significant.

## Figures and Tables

**Figure 1 molecules-24-02604-f001:**
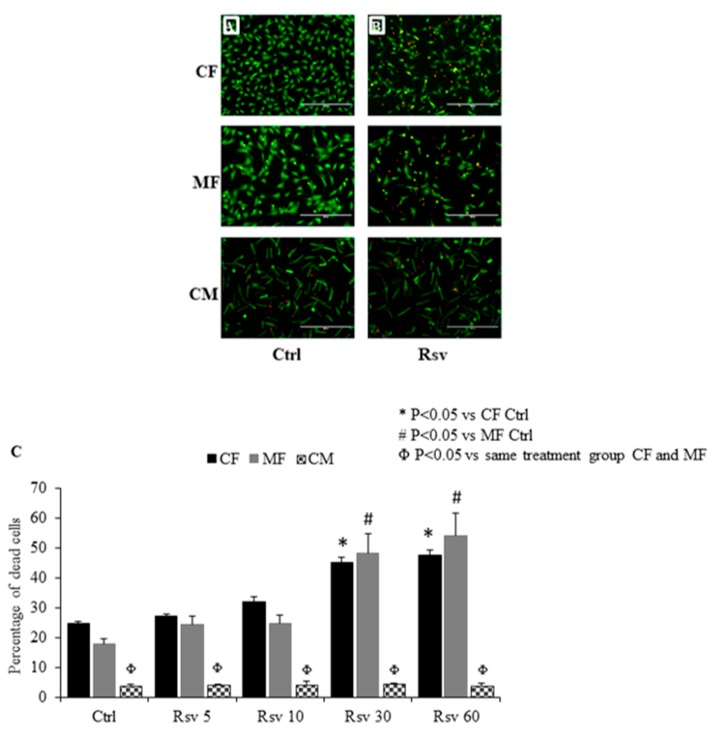
Dose dependent effect of resveratrol (Rsv) on cell survival of cardiac fibroblast (CF) myofibroblasts (MF) and cardiomyocytes (CM). Representative images of merged green and red fluorescence channels from calcein AM and ethidium homodimer 1 dyes are presented here. Cells appearing as yellow or red in the merged image are considered as dead cells. First row shows CF, second row shows MF and third row shows CM. Panel (**A**) represents control (Ctrl) and panel (**B**) represents Rsv 30 µM. (**C**) Graphical representation shows percentage of dead cells in CF, MF and CM in Ctrl and Rsv (5,10,30,60 µM) groups. *N* = 4. * *P* < 0.05 vs. CF Ctrl; # *P* < 0.05 vs. MF Ctrl; Φ *P* < 0.05 vs. same treatment group CF and MF.

**Figure 2 molecules-24-02604-f002:**
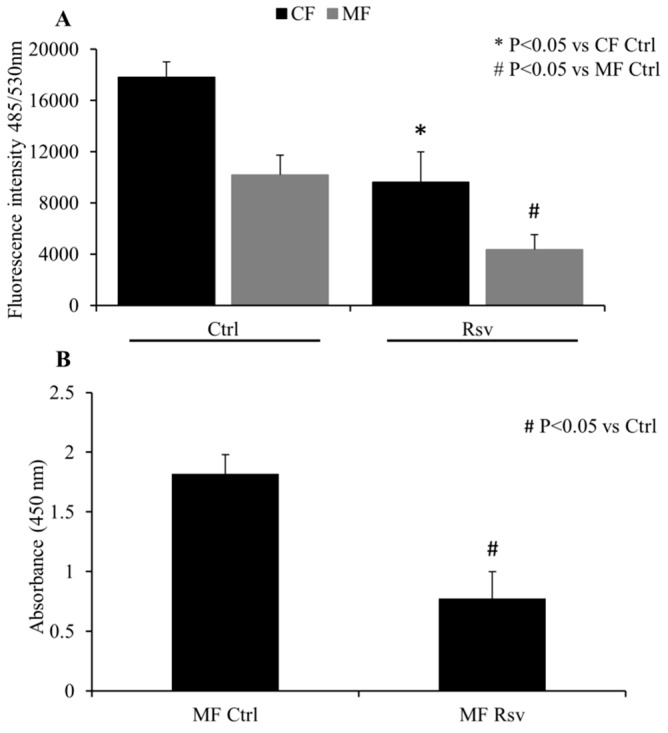
Effect of resveratrol (Rsv) on cardiac fibroblast (CF) and myofibroblast (MF) proliferation. (**A**) Relative cell count was measured as fluorescence intensity by spectrophotometer (excitation 485 nm/emission 530 nm) in CF and MF. *N* = 4. (**B**) MF proliferation measured by BrdU cell proliferation assay. N = 4. * *P* < 0.05 vs. CF Ctrl; # *P* < 0.05 vs. MF Ctrl.

**Figure 3 molecules-24-02604-f003:**
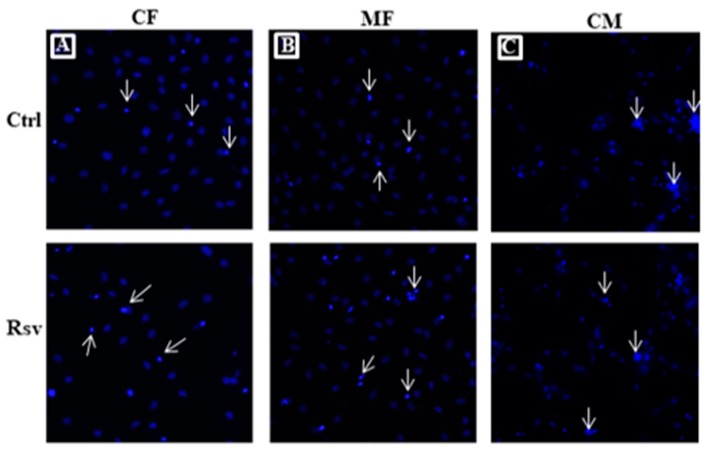
Effect of resveratrol (Rsv) in inducing apoptosis in cardiac fibroblast (CF) myofibroblasts (MF) and cardiomyocytes (CM). Representative images of CF, MF and CM with blue fluorescence from Hoechst dye. Arrows shows cells with condensed nuclei. Panel (**A**) represents control (Ctrl) and panel (**B**) represents Rsv (30 µM) treated groups. First row represents CF, second row represents MF and third row represents CM. (**C**,**D**) Graphical representation of percentage of CF, MF and CM with condensed nuclei. *N* = 4. * *P* < 0.05 vs. CF Ctrl; # *P* < 0.05 vs. MF Ctrl; Φ *P* < 0.05 vs. same treatment group CF; ¥ *P* < 0.05 vs. same treatment group MF.

**Figure 4 molecules-24-02604-f004:**
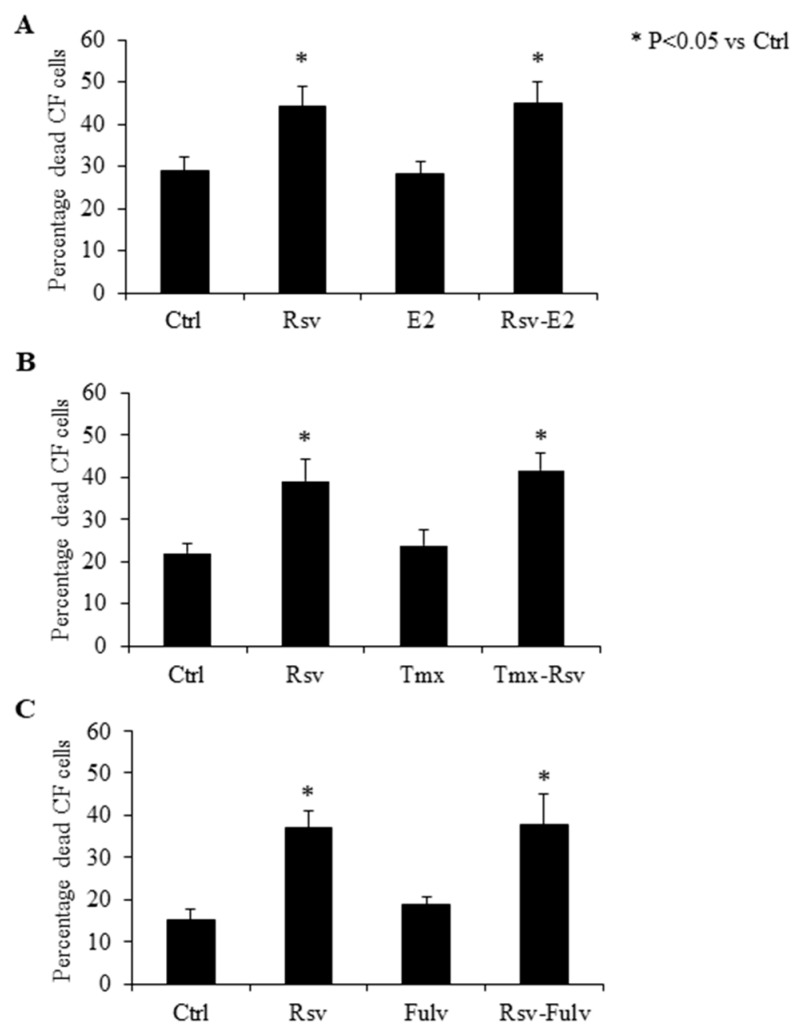
Resveratrol mediated cell death in cardiac fibroblasts (CF) is independent of estrogen receptor alpha signaling. Effect of (**A**) estrogen (E2), (**B**) tamoxifen (Tmx) and (**C**) fulvestrant (Fulv) on cell survival of cardiac fibroblasts (CF) treated with and without resveratrol (Rsv). Total percentages of dead cells were calculated from live and dead assay. *N* = 4–6. Control, Ctrl. * *P* < 0.05 vs. ctrl.

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
