# Peer review of "Divergent Effects of Resveratrol on Rat Cardiac Fibroblasts and Cardiomyocytes"

_molecules, 2019, doi:10.3390/molecules24142604_

Round 1

Reviewer 1 Report

This is an interesting study that further investigated some effects of the natural compound resveratrol in the context of cardiac disease especially fibrosis. The authors used a well-established cell culture model utilizing primary rat cardiac fibroblasts and cardiomyocytes. Experiments to study the mechanism of action of resveratrol are missing but these aims could be part of further studies as noted in the conclusion. Some minor points should to be corrected before publication.

1. abstract:

line 14: Don’t use „phytoestrogen“ because it is not clear if resveratrol directly can act as agonist or antagonist on the estrogen receptor. Better use, for example, „phytoalexin“.

Line 18: Avoid the use of two abbreviations for the same thing. Here, don’t use P0 and P1 but CF and MF throughout the manuscript (also line 122, 155).

Line 26: Please write: ….rat cardiac fibroblasts and myofibroblasts …

2. results:

line 129: To compare basal fluorescent intensity of MF and CF, it is necessary to plate the same amount of cells at the beginning of the experiment. I think this has been considered but you should mention it explicitly.

3. discussion:

line 223: replace RES by Rsv.

Reviewer 2 Report

the ms tested the protective effects of resveratrol on primary rat cardiac fibroblast an on cardiomyocites

instead of an interesting topic, the ms is quite confusing, with some experiments not clearly conducted. in particular: different timing for fibroblasts and cardiomyocites, an absence of real dose-dependent curve for both cell lines, not clarified involvment of estrogen signalling, materials and methods section not completely repeteable, discussion too long.

Reviewer 3 Report

This is a very interesting finding that has a lot of potential. More depth would be beneficial as would combining some of your figures.

Why were cardiomyocytes treated after 24 hours, but fibroblasts at 48? I understand the myofibroblasts, but not this. Time in culture can affect proliferation for many cells, so not sure how this would affect it. You also do not discuss seeding density, which may also influence your results. Are they seeded such that the density is similar at the time that Rsv is added?

If you intend to discuss differences in cell death due to Rsv treatment between the 3 groups then a 2-way ANOVA should be done with a post hoc t-test. These would also do well in one graph. 

Apoptosis - you only look at cell death and a late stage apoptotic marker. Consider looking at an early stage and/or mid stage marker as well. This is typical when confirming apoptosis.

Why did you choose tamoxifen? It can bind both ER receptors and sometimes has divergent effects on different cell types. Why not consider inhibitors specific to ERalpha and ERbeta?

Your figures could use some work. 

Figure 1 - Panels A-F don't add much value. Consider removing. Consider combining data into 1 graph.

Figure 2 &3 - why are these 2 separate figures? Your cell background seems high, especially for the red channel. Your dead stain doesn't look like actual staining and all appear red. How did you determine which were dead? Cells can express both if dying, but I don't think that is what is happening in these pictures. Maybe autofluorescence if your exposure is very high?

Figure 4 - Why did you not show cardiomyoctes as well?

Honestly, this is really nice information; however, there is no mechanism at all. More in depth analysis could make this a very nice paper.

Round 2

Reviewer 2 Report

the ms has been really improved after revision. But, I still have some concerns, in particular regarding the measurements of apoptotic cells only by means of  Hoechst assay, as well as the proliferation assay performed only by CyQUANT Direct Cell Proliferation Assay.

Author Response

Response to reviewer

Overall, we would like to thank the reviewer for their positive and constructive comments. Please find below our detailed, point-by-point response to the reviewer’s comments.

Reviewer 2

The ms has been really improved after revision. But, I still have some concerns, in particular regarding the measurements of apoptotic cells only by means of Hoechst assay, as well as the proliferation assay performed only by CyQUANT Direct Cell Proliferation Assay.

We have now quantified the proliferation of myofibroblasts (MF) with/without resveratrol treatment using a BrdU cell proliferation assay. We used MF for this study as we are able to accurately seed equal cell numbers (1x105 cells/well) of cells into each well. The results from this experiment are consistent with the earlier data presented in this manuscript that looked at cell growth following treatment of fibroblasts and myofibroblasts. Resveratrol significantly reduced proliferation of MF cells. The methods (page 6, paragraph 2, lines 134-144 and page 7, paragraph 1, lines 145-155), results (page 10, paragraph 2, lines 206-209) and discussion (page 15, paragraph 1, lines 290-293) sections have been revised accordingly in the new version of the manuscript. This data is now added as new figure (Figure 2b, Page 11) in the revised manuscript. We believe that the addition of this experiment has significantly strengthened our manuscript. Thank you for this very helpful suggestion

  Given the limited time for revisions, we did not conduct further studies to measure apoptosis following resveratrol treatment in these cells. However, we will use such assays in future studies to further validate the role of apoptosis. Thank you for this suggestion.  

Reviewer 3 Report

I am satisfied with most of the changes and only have one minor item that needs to be addressed regarding the proliferation assay. Please include this description as to how you assured similar numbers of cells. This information is important as similar numbers of cells per well are critical. You may want to note somewhere that it isn't homogeneous; however, fibroblasts proliferate much more rapidly and make up the majority of the proliferation.

Author Response

Response to reviewers

Overall, we would like to thank the reviewer for the positive and constructive comment. Please find below our response to the reviewer’s comment.

Reviewer 3

I am satisfied with most of the changes and only have one minor item that needs to be addressed regarding the proliferation assay. Please include this description as to how you assured similar numbers of cells. This information is important as similar numbers of cells per well are critical. You may want to note somewhere that it isn't homogeneous; however, fibroblasts proliferate much more rapidly and make up the majority of the proliferation.

Thank you for this helpful suggestion. We have now included a few sentences in the methods section (page 6, paragraph 2, lines 138-143) to clarify that cells were not counted before seeding for the CyQUANT Direct Cell Proliferation Assay due to the isolation procedures.

We have now quantified the proliferation of myofibroblasts (MF) with/without resveratrol treatment using a BrdU cell proliferation assay. We used MF for this study as we are able to accurately count and seed equal cell numbers (1x105 cells/well) of cells into each well. The results from this experiment are consistent with the earlier data presented in this manuscript that looked at cell growth following treatment of fibroblasts and myofibroblasts. Resveratrol significantly reduced proliferation of MF cells. The methods (page 6, paragraph 2, lines 134-144 and page 7, paragraph 1, lines 145-155), results (page 10, paragraph 2, lines 206-209) and discussion (page 15, paragraph 1, lines 290-293) sections have been revised accordingly in the new version of the manuscript. This data is now added as new figure (Figure 2b, Page 11) in the revised manuscript. We believe that the addition of this experiment has significantly strengthened our manuscript. Thank you for this very helpful suggestion